# Effectiveness of antimalarial drug combinations in treating concomitant urogenital schistosomiasis in malaria patients in Lambaréné, Gabon: A non-randomised event-monitoring study

Rella Zoleko-Manego[1,2,3], Dearie G. Okwu[1], Christian Handrich[1], Lia B. Dimessa-Mbadinga[1], Malick A. Akinosho[1], Wilfrid F. Ndzebe-Ndoumba[1], Saskia D. Davi[1,3], Daniel Stelzl[1], Luzia Veletzky[1,3,4], Andrea Kreidenweiss[1,2], Tamara Nordmann[3], Ayola A. Adegnika[1,2,5,6], Bertrand Lell[1,4], Peter G. Kremsner[1,2,5,6], Michael Ramharter[1,3,7], Ghyslain Mombo-Ngoma[1,5,7,8]*

1 Centre de Recherches Médicales de Lambaréné (CERMEL), Lambaréné, Gabon, 2 Institut für Tropenmedizin, Universität Tübingen, Tübingen, Germany, 3 Department of Tropical Medicine, Bernhard Nocht Institute for Tropical Medicine & I. Dep. of Medicine, University Medical Center Hamburg-Eppendorf, Hamburg, Germany, 4 Division of Infectious Diseases and Tropical Medicine, Department of Medicine 1, Medical University of Vienna, Vienna, Austria, 5 German Center for Infection Research (DZIF), African partner institution, CERMEL, Gabon, 6 German Center for Infection Research (DZIF), partner site Tübingen, Germany, 7 German Center for Infection Research (DZIF), partner site Hamburg-Lübeck-Borstel-Riems, Germany, 8 Department of Implementation Research, Bernhard Nocht Institute for Tropical Medicine & I. Dep. of Medicine, University Medical Center Hamburg-Eppendorf, Hamburg, Germany

* ghyslain.mombo-ngoma@bnitm.de

## Abstract

### Background

Urogenital schistosomiasis is prevalent in many malaria endemic regions of sub-Saharan Africa and can lead to long-term health consequences if untreated. Antimalarial drugs used to treat uncomplicated malaria have shown to exert some activity against *Schistosoma haematobium*. Here, we explore the efficacy on concomitant urogenital schistosomiasis of first-line recommended artemisinin-based combination therapies (ACTs) and investigational second-generation ACTs when administered for the treatment of uncomplicated malaria in Gabon.

### Methods

Microscopic determination of urogenital schistosomiasis was performed from urine samples collected from patients with confirmed uncomplicated malaria. Egg excretion reduction rate and cure rate were determined at 4-weeks and 6-weeks post-treatment with either artesunate-pyronaridine, artemether-lumefantrine, artesunate-amodiaquine or artefenomel-ferroquine.

### Results

Fifty-two (16%) out of 322 malaria patients were co-infected with urogenital schistosomiasis and were treated with antimalarial drug combinations. *Schistosoma haematobium* egg

**Data Availability Statement:** All relevant data are within the manuscript and its Supporting Information files.

**Funding:** The author(s) received no specific funding for this work.

**Competing interests:** The authors have declared that no competing interests exist.

excretion rates showed a median reduction of 100% (interquartile range (IQR), 17% to 100%) and 65% (IQR, -133% to 100%) at 4-weeks and 6-weeks post-treatment, respectively, in the artesunate-pyronaridine group (n = 20) compared to 35% (IQR, −250% to 70%) and 65% (IQR, -65% to 79%) in the artemether-lumefantrine group (n = 18). Artesunate-amodiaquine (n = 2) and artefenomel-ferroquine combination (n = 3) were not able to reduce the rate of eggs excreted in this limited number of patients. In addition, cure rates were 56% and 37% at 4- and 6-weeks post-treatment, respectively, with artesunate-pyronaridine and no cases of cure were observed for the other antimalarial combinations.

## Conclusions

Antimalarial treatments with artesunate-pyronaridine and artemether-lumefantrine reduced the excretion of *S. haematobium* eggs, comforting the hypothesis that antimalarial drugs could play a role in the control of schistosomiasis.

## Trial Registration

This trial is registered with clinicaltrials.gov, under the Identifier NCT04264130.

### Author summary

This study aimed to evaluate the effects on schistosomiasis, of antimalarial drugs given to treat malaria in adults and children. The question is relevant as malaria and schistosomiasis co-infect patients in endemic areas. There is no systematic screening of schistosomiasis and some patients may hide it because of stigmatisation. On the other hand, malaria treatment is very frequently given in these areas sometimes even without any biological confirmation. This is a proof of concept study that shows some evidence of efficacy of antimalarial drugs, particularly artesunate-pyronaridine and to some lesser extent artemether-lumefantrine on schistosomiasis. These findings if confirmed in larger and well-designed prospective studies highlight the potential double benefit of controlling and managing both parasitic infections simultaneously.

## Introduction

Schistosomiasis is among the most important water-borne diseases and it is globally the second most important parasitic infection after malaria in tropical and subtropical areas of Africa [1,2]. According to the World Health Organization (WHO), it is a disease that affects 230 million people worldwide, of which 90% of cases live in Africa [3]. Co-endemicity of schistosomiasis and malaria is common in many Sub-Saharan African settings including rural Gabon, affecting preferentially poor rural populations [4,5]. In Gabon, transmission of malaria is perennial with little variation during dry and rainy seasons [6]. Limited access to safe drinking water and adequate sanitation in some rural and semi-urban areas of Gabon are reasons for perennial transmission of schistosomiasis. Thus, most communities in these areas are at continued risk of contracting and experiencing the morbidity and mortality associated with both diseases. School aged children and young adults are at highest risk of schistosomiasis with haematuria as typical signs.

Praziquantel (PZQ) remains the only drug recommended by WHO for the treatment of schistosomiasis. PZQ is safe and effective against adult schistosome worms [7,8], but ineffective against juvenile developmental stages [9,10]. Contrary to this, some antiparasitic drugs, notably artemisinin derivatives were found to be active on schistosomula, suggesting their use in combination with PZQ for an effective treatment of the disease or for prevention [11].

Artemisinin-based combination therapy (ACT) has made a major contribution to reduce the global malaria burden and is currently the first-line treatment for uncomplicated malaria. The two artemisinin derivatives artemether and artesunate have shown activity against schistosomiasis [9,12–17]. In addition, mefloquine and other antimalarial drugs have demonstrated clinically significant activity against *Schistosoma haematobium* [18]. However, the usefulness of these drugs for schistosomiasis treatment or prevention is debated for their overall activity and their potential to contribute to the development of drug resistance of the malaria parasite. In Gabon, low efficacy of artesunate monotherapy on urogenital schistosomiasis have been reported by Bormann et *al.* [19] and the activity of artemether has not yet been assessed. Artemether-lumefantrine, artesunate-amodiaquine, and artesunate-pyronaridine are three ACTs registered and available in Gabon. To further characterize the effect of antimalarial combination therapy on urogenital schistosomiasis, we assessed the efficacy of available ACTs and investigational antimalarial combination therapies on concomitant urogenital schistosomiasis when administered for the treatment of uncomplicated malaria in a region where *S. hematobium* and *Plasmodium falciparum* are the main prevalent species of schistosomiasis and malaria.

## Methods

### Ethical statement

The study was approved by the institutional ethics committee of CERMEL, reference number: CEI- 006/2018. The study was conducted in line with Good Clinical Practice (GCP) principles of the International Conference on Harmonization (ICH) and the Declaration of Helsinki as well as the local and national guidelines. All participants or their parents or guardians signed a written informed consent prior to any study specific procedure. The study was registered at clinicaltrials.gov registry under the Identifier NCT04264130.

### Study design and population

The present study is a non-randomized pre-test and post-test study with a prospective follow-up of adult and paediatric patients treated for uncomplicated malaria to assess the effectiveness of antimalarial combination therapy on concomitant urogenital schistosomiasis. The study was carried out by the Centre de Recherches Médicales de Lambaréné (CERMEL) [20], in Lambaréné and the surrounding area in Gabon between August 2018 and November 2019. Lambaréné is located in the central region of Gabon within the equatorial rainforest and is highly endemic for malaria and urogenital schistosomiasis [5,21,22].The region is characterised by the Ogooué river and its tributaries, with numerous ponds, lakes and streams representing laundry and bathing sites but also important points of contact with *S. haematobium* [5,23].

Study participants were patients of both sexes, of all ages, with uncomplicated malaria treated with antimalarial combination therapy, and for whom a urine sample collected prior to treatment (pre-test) was positive for *S. haematobium* and with no intake of praziquantel during the six previous weeks. For all participants a written informed consent was obtained, as well as acceptance for follow up until six weeks post-treatment of malaria (post-test). Patients were actively followed up during the 6-week period and were treated with standard PZQ therapy at

the end of the follow-up period. Anti-schistosomal therapy was deferred to avoid unknown drug-drug interaction with investigational antimalarial drugs.

A copy of the protocol is provided in the Supporting information (S1 Study Protocol). This study is reported as per the non-randomized clinical trial (Trend) guideline (S1 TREND Checklist). The trial was conducted according to protocol and no changes to the methods were made after trial commencement.

## Sample size determination

We aimed to descriptively assess the effect of ACTs on *S. haematobium* egg excretion in the context of clinical trials for antimalarial drugs. Due to the descriptive nature of the study no statistical hypothesis testing was defined. In this present analysis, all participants with *S. haematobium* egg excretion at baseline and assessment four and six weeks after ACT administration were included.

## Study procedures

At baseline, socio-demographic data were recorded, and participants were tested for urinary excretion of *schistosoma* eggs (pre-test). All patients were either participants of antimalarial drug trials or were offered first-line standard treatment of malaria with either artemether-lumefantrine or artesunate-amodiaquine. During the study period, clinical trials with artesunate-pyronaridine and artefenomel-ferroquine were ongoing (study ID number: SP-C-021-15, ClinicalTrials.gov Identifier: NCT03201770; DRI12805, ClinicalTrials.gov Identifier: NCT02497612, and ACT14655, ClinicalTrials.gov Identifier: NCT03660839, respectively). Artesunate-pyronaridine, artesunate-amodiaquine and artemether-lumefantrine are classical artemisinin-based combination therapies. Artefenomel-ferroquine may be considered as a second-generation synthetic artemisinin-based combination therapy as artefenomel is a synthetic trioxolane closely related to the artemisinin class of antimalarials, and ferroquine which is a representative of second generation 4-aminoquinolines as partner drug.

Malaria treatment was administered in accordance with the label information or the specific study protocol. Study medication was administered orally as follow:

Fixed-dose artesunate–pyronaridine (Shin Poong Pharmaceutical, Seoul, South Korea) was administered according to body weight once daily for 3 days; for tablet (180:60 mg), one tablet ($\geq$20 to <24 kg), two tablets ($\geq$24 to <45 kg), three tablets ($\geq$45 to <65 kg), or four tablets ($\geq$65 kg); and for granule sachets (60:20 mg), one sachet ($\geq$5 to <8 kg), two sachets ($\geq$8 to <15 kg), or three sachets ($\geq$15 to <20 kg).

Fixed-dose artemether–lumefantrine (Coartem, Novartis, Basel, Switzerland) was administered twice daily for 3 days as one tablet of 20/120 mg for body weights 5 to <15 kg, two tablets for weights $\geq$15 to <25 kg, or three tablets for weights of $\geq$25 to <35 kg, and one table of 80/480 mg for body weight $\geq$ 35kg.

Fixed-dose artesunate–amodiaquine (ASAQ Winthrop, Sanofi, Paris, France) was administered once daily for 3 days as two tablets of 100 mg /270 mg for body weights $\geq$36 kg.

Artefenomel (OZ439) and ferroquine (Sanofi, Paris, France) were a single dose administration of artefenomel 300mg and ferroquine 400mg for subject involved in ACT14655 study and a single dose of artefenomel 800mg and four different doses of ferroquine (400, 600, 800, 1200 mg) adapted to body weight if weight < 35kg for subject randomised in DRI12850 Study.

Four and six weeks after malaria treatment coinciding with patients' visits in the antimalarial drug trials, urine samples were collected during two consecutive days to assess participants' eggs excretion status (post-test). On the day of collection, all effort was done to collect urine between 10:00 and 15:00 as urinary excretion of eggs follows a daily rhythm with a peak

around noon. No physical exercise prior to urine collection was advised due to the acute malaria episode.

All participants received at the last follow-up visit a single dose of 40 mg/kg of PZQ for the treatment of schistosomiasis, as recommended by national treatment guidelines, with no further follow-up.

## Laboratory procedures

Urine test strip and urine filtration technique were performed on each urine sample. A urine test strip (Combur-10-Test, Roche Diagnostic, Cobas) was performed to assess the presence of haematuria. The lower detection limit of blood (erythrocyte) was 5 erythrocytes/µl. For the diagnosis of urogenital schistosomiasis, urine samples were subjected to the filtration technique and read by microscopists unaware of study treatment. Urine filtration technique consisted of 10 mL of urine freshly collected injected through a chamber containing a 12µm pore-size nuclepore filter followed by microscopic examination for the detection of *S. haematobium* eggs as described elsewhere [24,25]. The counting of eggs is done using x40 magnification scanning completely the filter in four passes and the egg count per volume of urine is recorded. Samples were considered *S. haematobium* positive if at least one egg was detected in at least one urine sample out of at least two urine samples provided by each subject and as negative if all two or more urine samples did not have any eggs present.

## Data analysis

Data were captured in patients forms and entered in a REDCap database hosted at CERMEL [26]. The clean database was imported into Stata version 13.1 (StataCorp, College Station, TX) for statistical analysis. Qualitative data were summarized as number and percentage, while quantitative data were presented as median and interquartile range (IQR). *Schistosoma haematobium* infection intensity was classified as light when eggs count per 10ml of urine was less than 50 and heavy when it was equal to or more than 50 according to WHO criteria [27]. The primary endpoint was the egg excretion reduction (EER) assessed 6-weeks after initiation of antimalarial treatment with an ACT. The secondary endpoints included EER at 4-weeks post-treatment and cure rate (CR) at 4- and 6-weeks post-treatment calculated as the percentage of volunteers cured among those treated, and proportions of subjects with micro-haematuria before and after treatment. Because of a higher proportion of heavy schistosomiasis infection in artemether–lumefantrine treatment group compared to the artesunate–pyronaridine treatment group, one sub-analysis was performed to explore whether there was any alteration that could be imputed to the infection intensity. A second sub-analysis comparing baseline characteristics of subject who had cleared their eggs and those who still had eggs counted during follow-up period was performed in a post-hoc analysis. The graphical depiction of the evolution of schistosoma eggs following initiation of antimalarial treatment was drawn using GraphPad Prism 9.0.0 (GraphPad Software Inc.).

## Results

### Study population and baseline characteristics

A total of 351 patients with confirmed malaria gave their consent to be screened for urinary schistosomiasis and to be followed-up for up to six weeks if positive. The details of participants' flow are depicted in Fig 1. Of the 321 who provided urine samples, 52 (16%) were positive for *S. haematobium*. Of the *Schistosoma*-positive patients, 24 were treated with artesunate–pyronaridine, 23 with artemether–lumefantrine, two with artesunate–amodiaquine, and three with

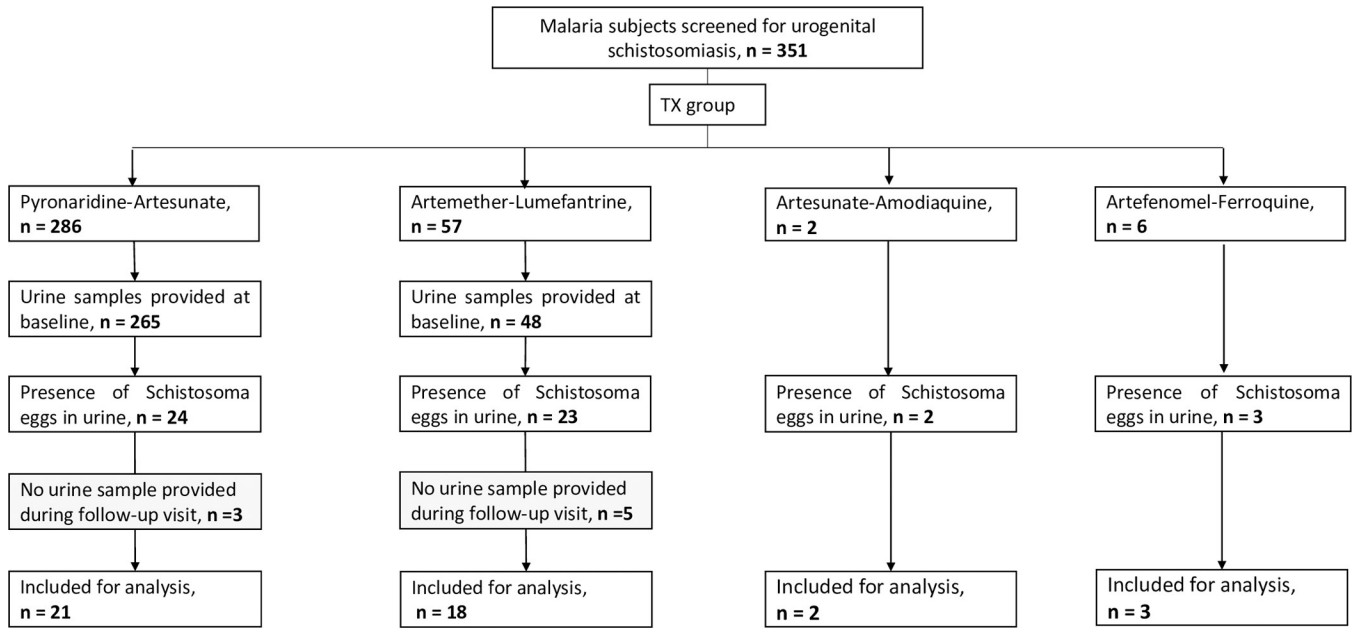

**Fig 1. Participants flow.** Abbreviation: Tx: treatment.

artefenomel–ferroquine. Eight patients who did not provide any urine samples during the follow-up visits were excluded from the analysis (Fig 1).

A total of 39 out of 47 patients treated with either artesunate–pyronaridine or artemether–lumefantrine completed at least one follow up visit. The overall median age was 11.7 (IQR: 7.2–15.9) years. Most patients, 59.0% (23/39) were school age children and there were more males than females (Table 1).

**Table 1. Baseline characteristics of study participants treated with artesunate-pyronaridine and artemether-lumefantrine and included in the analysis, N = 39.**

| Characteristics | Overall study population | AP | AL |
|---|---|---|---|
| **N (%)** | 39 (100.0) | 21(53.8) | 18 (46.2) |
| **Age in years, median (IQR)** | 11.7 (7.2–15.9) | 13.6 (9.6–17.9) | 8.5 (6.7–13.6) |
| **School age children (<14 years), n (%)** | 23 (59.0) | 10 (47.6) | 13 (72.2) |
| **Adolescents and adults (≥14 year), n (%)** | 16 (41.0) | 11 (52.4) | 5 (27.8) |
| **Sex** | | | |
| Female, n (%) | 11 (28.2) | 5 (23.8) | 6 (33.3) |
| Male, n (%) | 28 (71.8) | 16 (76.2) | 12 (66.7) |
| **Median egg count /10ml (IQR)** | 55(10–431) | 30 (6.0–103) | 105 (39–1293) |
| **Infection intensity** | | | |
| Light infection (< 50 eggs/10ml), n (%) | 18 (46.2) | 13 (61.9) | 5 (27.8) |
| Heavy infection (≥ 50 eggs/10ml), n (%) | 21 (53.8) | 8 (38.1) | 13 (72.2) |
| **Hematuria, n (%)** | 36 (92.3) | 18 (87.7) | 18 (100) |
| **Location** | | | |
| Lambaréne, n (%) | 33 (84.6) | 16 (76.2) | 17 (94.4) |
| Vicinity, n (%) | 6 (15.4) | 5 (23.8) | 1 (5.6) |

Abbreviations: AP: artesunate-pyronaridine; AL: artemether-lumefantrine; N or n: number, IQR: interquartile range

The median egg load at baseline was 55 (IQR: 10–431), with lower loads in the artesunate–pyronaridine group compared to the other treatment groups, and there were more light infections in the artesunate–pyronaridine group while there were more heavy infections in the other treatment groups, as shown in Table 1. At baseline, almost all patients had blood in urine as detected by urinalysis. There was no difference in location of residence of patients in the artemether–lumefantrine and artesunate–pyronaridine groups, respectively (94.4% vs 76.2%).

Two adolescent, male and female were treated with artesunate–amodiaquine. Both had blood detected in urine with egg count of 1 and 251 eggs/10ml of urine respectively at baseline. Two school aged children and one adult, all of them female, received artefenomel–ferroquine as antimalarial drug with the median egg count of 124 (46–238) eggs/10 ml of urine at baseline.

## Egg excretion reduction and cure rate

For the artesunate–pyronaridine treatment group, 16 and 19 patients were evaluated at 4-weeks and 6-weeks post-treatment, respectively. The egg excretion reduction was 100% (IQR; 17–100) at 4-weeks and 65% (IQR; -133–100) at 6-weeks post-treatment. Nine patients (56.3%) out of 16 at 4-weeks and seven patients (36.8%) out of 19 at 6 weeks, were classified as cured as they did not excrete any egg in any of the samples provided during two consecutive days. Seven and 12 patients did not achieve parasitological cure at 4- and 6-weeks post-treatment, respectively. Median egg excretion was 37 (IQR; 20–598) eggs/10 ml of urine with an egg excretion reduction at -10% (IQR, -633- 90) at 4-weeks, while median egg excretion was 87 (IQR; 12–379) eggs/10ml of urine with egg excretion reduction at -35% (IQR; -590–59) at 6-weeks. In this treatment group, haematuria was observed in 9 (56.3%) out of 16 and in 12 (66.7%) out of 18, at 4 weeks and 6 weeks post-treatment, respectively, compared to 83.3% (20/24) at baseline (Table 2).

**Table 2. Schistosoma egg excretion reduction and cure rate post malaria treatment.**

|  | AP | | AL | |
|---|---|---|---|---|
|  | **Median** | **Range** | **Median** | **Range** |
| **4-Weeks post-treatment** |  |  |  |  |
| N | 16 | | 15 | |
| Baseline[a] Egg excretion/10ml | 17 | 6–76 | 120 | 36–1704 |
| Egg excretion/10ml | 0 | 0–30 | 164 | 62–500 |
| Egg excretion reduction, % | 100 | 17–100 | 35 | -250–70 |
| Cure rate, n (%) | 9 (56) |  | - |  |
| Hematuria, n (%) | 9 (56) |  | 13** (100) |  |
| **6-Weeks post-treatment** |  |  |  |  |
| N | 19 | | 18 | |
| Baseline Egg excretion/10ml | 19 | 5–103 | 105 | 39–1293 |
| Egg excretion/10ml | 9 | 0–304 | 56 | 14–209 |
| Egg excretion reduction, % | 65 | -133–100 | 65 | -65–79 |
| Cure rate, n (%) | 7 (37) |  | - | - |
| Hematuria, n (%) | 12* (67) |  | 18 (100) |  |

Abbreviations: AP: artesunate-pyronaridine; AL: artemether-lumefantrine; ASAQ: artesunate-amodiaquine; OZFQ: artefenomel (OZ439)-ferroquine

*: 1 missing data

**: 2 missing data; N: total number of participants

For the artemether–lumefantrine treatment group, 15 and 18 patients were evaluated at 4-weeks and 6-weeks post-treatment, respectively. There was a decrease in egg excretion from 517 eggs/10ml of urine at baseline to 164 eggs/10ml of urine at 4-weeks post-treatment. More-over, there was a reduction in egg excretion of 64% (IQR: -91–79) after 6-weeks of follow-up. None of the patients in that treatment group cleared their eggs to be classified as cured. At each follow-up visit, all patients evaluated presented with haematuria.

For the artesunate–amodiaquine group, two patients were evaluated at 4-weeks and one at 6-weeks post-treatment. There was a decrease in egg excreted at 4-weeks from 261 eggs/10ml to 61 eggs/10ml followed by an increase at 6-weeks post-treatment. No cure was observed and they continued to have haematuria.

For the artefenomel–ferroquine treatment group, two and three patients were evaluated at 4-weeks and 6-weeks post-treatment, respectively. There was an increase in egg excretion at 4-weeks post-treatment (-98% [IQR: -235.1 to 39]) and at 6-weeks post-treatment (-61% [IQR: -65 to -61]). None of the patients were cured and they all had haematuria present at follow-up.

## Analysis of schistosoma infection intensity subpopulation

33 participants were from Lambaréné among which 19 with heavy infection (Table 1). Out of the 21 participant with heavy infection, 12 (67%) were treated with artemether–lumefantrine.

Taking into account schistosoma infection intensity, egg excretion rate at 4-weeks was higher in artesunate–pyronaridine (97% [IQR: 91 to 100]) group compared to artemether–lumefantrine group (51% [IQR: -29 to 86]) on subjects with heavy schistosoma infection (Table 3). Egg excretion rate was at 70% (IQR: 52–87) in subjects with heavy schistosoma infection at 6-weeks posttreatment while it was at 69% (IQR: -43-96) in the artesunate–pyro-naridine group.

As depicted in Fig 2, artemether–lumefantrine was more effective 6-weeks post-treatment in general on each subject compared to artesunate–pyronaridine.

## Analysis of the artesunate-pyronaridine subpopulation

Overall, of the 21 patients followed-up in the artesunate–pyronaridine group, five (25%) were found with no eggs in urine at both 4- and 6-weeks post-treatment, while 7 (35%) still had eggs throughout the follow-up. The baseline median egg excretion was 5 eggs/10ml urine and 40 eggs/10ml urine in the cured and the non-cured groups, respectively (Table 4).

## Discussion

Our findings show that coinfection of malaria and schistosomiasis is common in the study area and that when administering ACTs for the treatment of malaria, there is an effect on schistosomiasis as assessed by urinary eggs excretion. These findings serve two purposes, one is to control two parasites with one treatment, meaning to have drugs that are effective at the same time against malaria and schistosomiasis. Secondly, the possible repurposing of antima-larial drugs to support the only anti-schistosomal drug praziquantel may become more relevant.

In our study population, the prevalence of urogenital schistosomiasis among malaria patients was 16%. This is in line with the 11% prevalence previously reported in the same region in school age children [28] or the 15% in pregnant women in Cameroon [29]. However, since the diagnostic method used was microscopy and there are other methods more sensitive including molecular diagnostic techniques, it is possible that the real burden of schistosomiasis and malaria coinfection is underestimated here. The knowledge of this high frequency of coin-fections of malaria and schistosomiasis is important as it may give a first idea of the collateral

**Table 3. Schistosoma infection intensity and schistosoma egg excretion during follow-up.**

| Infection intensity | AP | | | | | | AL | | | | | |
| --- | --- | --- | --- | --- | --- | --- | --- | --- | --- | --- | --- | --- |
| | 4-Weeks post-treatment | | | 6-Weeks post-treatment | | | 4-Weeks post-treatment | | | 6-Weeks post-treatment | | |
| | Baseline Egg excretion/ 10ml | Egg excretion/ 10ml | Egg excretion reduction % | Baseline Egg excretion/ 10ml | Egg excretion/ 10ml | Egg excretion reduction % | Baseline Egg excretion/ 10ml | Egg excretion/ 10ml | Egg excretion reduction% | Baseline Egg excretion/ 10ml | Egg excretion/ 10ml | Egg excretion reduction % |
| Light infection ($<$ 50 eggs/ 10ml), Median (IQR) | 6 (4–19) | 0 (0–37) | 100 (-23-100) | 6 (4–19) | 4 (0–38) | 53 (-13-100) | 17 (9–36) | 62 (21–183) | -250 (-370--72) | 17 (9–36) | 24 (14–27) | -13 (-167-25) |
| Heavy infection ($\geq$ 50 eggs/ 10ml), Median (IQR) | 235 (98–688) | 2 (0–20) | 97 (91–100) | 235 (103–322) | 160 (14–368) | 69 (-43-96) | 517 (89–1704) | 251 (130–565) | 51 (-29-86) | 517 (89–1704) | 133 (25–526) | 70 (52–87) |

Abbreviations: AP: artesunate-pyronaridine; AL: artemether-lumefantrine; IQR: interquantile range

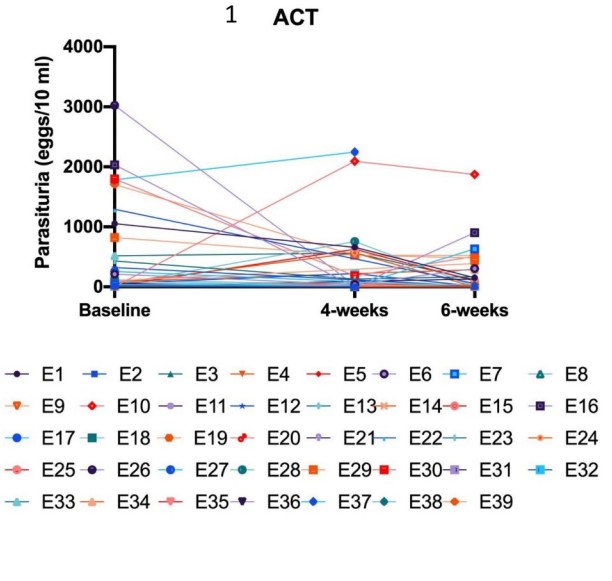

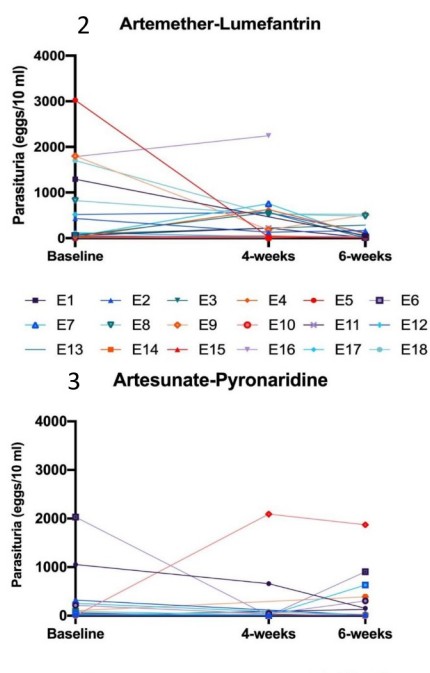

**Fig 2. Schistosoma egg excression before and posttreatment.**

benefit of the treatment of malaria as it may also treat an undiagnosed schistosomiasis. Indeed, for several reasons, schistosomiasis may be left unnoticed for prolonged periods of time due to cultural considerations of perception of haematuria being affected with a shame of talking about this symptom. Importantly, haematuria is also a symptom which is not "loud" compared to high fevers and other signs and symptoms of malaria and there are currently no systematic screening and prevention campaigns for schistosomiasis in Central Africa.

The activity of artemisinin derivatives against schistosomiasis has previously been reported, notably its efficacy on schistosomulae [30]. In addition, aminoquinoline antimalarials such as mefloquine have been shown to exert significant activity against *S. haematobium* [18]. Artesunate-pyronaridine and artemether-lumefantrine show a considerable effect after six weeks of

**Table 4. Baseline characteristics of cured and non-cured participants treated with artesunate-pyronaridine.**

| | | Subjects with eggs present during FU | Subjects with egg clear at D42 | Subjects with egg clear at D28 | Subjects with egg clear at D28 & D42 |
|---|---|---|---|---|---|
| | | n = 7 | n = 7 | n = 9 | n = 5 |
| **Age (median, IQR)** | | 14.6 (13.6–20.7) | 17.4; (9.6–22.5) | 11.7; (9.6–17.4) | 11.6; (9.6–17.4) |
| **Sex** | Male; n (%) | 6 (85.7) | 5 (71.4) | 6 (66.7) | 3 (60.0) |
| | Female; n (%) | 1 (14.3) | 2 (28.6) | 3 (33.3) | 2 (40.0) |
| **Residence** | Lambarene; n (%) | 3 (42.9) | 6 (85.7) | 7 (77.8) | 4 (80.0) |
| | Vinicity; n (%) | 4 (57.1) | 1 (14.3) | 2 (22.2) | 1 (20.0) |
| **Median egg count /10ml (IQR)** | | 40; (4.0–213.0) | 6.0; (4.0–30.0) | 6.0; (5.0–19.0) | 5.0; (4.0–6.0) |

2 patients who cleared eggs at 4-weeks missed visit at 6-weeks; 2 patients cleared eggs only at 6 weeks; 2 patients who cleared eggs at 4 weeks had reappearance of eggs at 6 weeks; IQR: interquartile range; FU: follow-up; n: number

follow-up with a median egg excretion reduction rate of 65% each. Indeed, sub-analysis of schistosoma infection intensity support these results. Moreover, in the artesunate-pyronaridine treatment group, there were patients apparently cured with no more excretion of eggs during the follow-up. These findings of a better effect of the artesunate-pyronaridine combination may indicate that although artemether and artesunate are closely related derivatives of artemisinin and are quickly metabolized to dihydroartemisinin in the human host and are thus mostly likely equally active against schistosomula, pyronaridine might exert some additional effect against schistosomes leading to a synergy of this combination. Nevertheless, this interpretation must be taken with caution as other factors such as older age associated with lower levels of infection in the pyronaridine-artesunate group may have also influenced the outcomes of our analysis. When looking within the artesunate-pyronaridine group, the characteristics of those cured were not different from those not cured except for the baseline egg load. Indeed, the five patients cured had a very light density of eggs excreted at baseline. Our results show that subjects with light infection intensity have a greater egg reduction rate compared to those with heavy infection intensity. This corroborates the result of Keiser *et al.* and Boulanger *et al.* [12,31] who reported that subjects with light infection are more likely to clear their parasites or to have a higher egg reduction rate. Importantly, we report elsewhere that pyronaridine showed activity against schistosomula and adult worms when tested in vitro and in mice [32]–a finding further substantiating our clinical findings of this study.

The artesunate-amodiaquine and artefenomel-ferroquine treatment groups did not contribute a lot to this analysis as the number of patients was very limited. Further data are therefore necessary to appreciate their anti-schistosomal potential *in vivo*. This study is limited by its design as it was not designed as a prospective interventional clinical trial but rather as an observational study. Hence, our study is a secondary analysis on data from malaria clinical trials designed to assess the effect of antimalarial drug combinations in patients with uncomplicated malaria. This has led to selection bias within groups with differences in the baseline characteristics as observed with the baseline egg excretion. The overall number of participants of this observational study is limited and a proportion of patients did not completely adhere to follow up schedule. There is a lack of knowledge about the effect of malaria itself on schistosomiasis egg excretion mainly due to the fact that malaria constitutes an acute illness resulting in symptoms that requires prompt treatment. Further research is needed to understand whether malaria may directly influence schistosomiasis egg excretion. Other limitations are the lack of robust and scalable read-outs of schistosome eggs viability after drug exposure and absence of some physical exercise before collecting the urine samples. Other laboratory techniques like detection of specific antigens such as circulating anionic antigen (CAA) and the circulating cathodic antigen (CCA) could help to understand the stage of activity whether on juvenile forms or adult worms.

Repurposing antimalarial drugs for the treatment of schistosomiasis either concomitantly or specifically for schistosomiasis will also pose the problem of the appropriate therapeutic regimen as the best potency is shown at higher doses and for longer durations than the usual 3-day period recommended for malaria treatment in *in vitro* assessments. However, despite these limitations this study provides first evidence for clinically important *in vivo* activity of these artemisinin-based combination therapies on urogenital schistosomiasis when administered for three days to treat uncomplicated malaria.

## Conclusion

Schistosomiasis and malaria coinfections are common in Gabon. Our findings provide first evidence that artemisinin-based combination therapies including artesunate-pyronaridine and

artemether-lumefantrine are effective in reducing egg excretion within four to six weeks following the treatment of uncomplicated malaria. These findings suggest that there could be some collateral benefit of these antimalarial regimens in treating malaria in regions where both malaria and schistosomiasis are co-endemic. Further studies, larger in size and with prospective interventional design are needed to further characterize that effect in humans.

## Supporting information

**S1 TREND Checklist. TREND Checklist.**
(DOC)

**S1 Database. Suppelmentary data.**
(XLSX)

**S1 Study Protocol. Study Protocol.**
(PDF)

## Acknowledgments

We are thankful to the patients who accepted to take part to this study. We are grateful to the staff of the department of Clinical Operations at the Centre de Recherches Médicales de Lambaréné, particularly Ms. Hornella Ekoume Ngombibang and Ms. Carlene Willia Bouanga Mombo.

## Author Contributions

**Conceptualization:** Rella Zoleko-Manego, Ayola A. Adegnika, Bertrand Lell, Peter G. Kremsner, Michael Ramharter, Ghyslain Mombo-Ngoma.

**Data curation:** Rella Zoleko-Manego, Dearie G. Okwu, Christian Handrich, Lia B. Dimessa-Mbadinga, Malick A. Akinosho, Wilfrid F. Ndzebe-Ndoumba, Saskia D. Davi, Daniel Stelzl, Tamara Nordmann.

**Formal analysis:** Rella Zoleko-Manego, Tamara Nordmann, Michael Ramharter, Ghyslain Mombo-Ngoma.

**Investigation:** Rella Zoleko-Manego, Dearie G. Okwu, Christian Handrich, Lia B. Dimessa-Mbadinga, Malick A. Akinosho, Wilfrid F. Ndzebe-Ndoumba, Saskia D. Davi, Daniel Stelzl, Luzia Veletzky, Andrea Kreidenweiss, Tamara Nordmann, Ayola A. Adegnika, Bertrand Lell.

**Methodology:** Rella Zoleko-Manego, Christian Handrich, Luzia Veletzky, Andrea Kreidenweiss, Ayola A. Adegnika, Peter G. Kremsner, Michael Ramharter, Ghyslain Mombo-Ngoma.

**Project administration:** Rella Zoleko-Manego, Ayola A. Adegnika, Bertrand Lell, Peter G. Kremsner, Michael Ramharter, Ghyslain Mombo-Ngoma.

**Resources:** Dearie G. Okwu, Saskia D. Davi, Bertrand Lell, Peter G. Kremsner, Ghyslain Mombo-Ngoma.

**Software:** Tamara Nordmann, Ayola A. Adegnika, Bertrand Lell, Peter G. Kremsner, Michael Ramharter, Ghyslain Mombo-Ngoma.

**Supervision:** Rella Zoleko-Manego, Dearie G. Okwu, Christian Handrich, Lia B. Dimessa-Mbadinga, Malick A. Akinosho, Wilfrid F. Ndzebe-Ndoumba, Saskia D. Davi, Daniel Stelzl, Peter G. Kremsner, Michael Ramharter, Ghyslain Mombo-Ngoma.

**Validation:** Rella Zoleko-Manego, Dearie G. Okwu, Christian Handrich, Lia B. Dimessa-Mbadinga, Malick A. Akinosho, Wilfrid F. Ndzebe-Ndoumba, Saskia D. Davi, Daniel Stelzl, Tamara Nordmann.

**Writing – original draft:** Rella Zoleko-Manego, Ghyslain Mombo-Ngoma.

**Writing – review & editing:** Rella Zoleko-Manego, Dearie G. Okwu, Christian Handrich, Lia B. Dimessa-Mbadinga, Malick A. Akinosho, Wilfrid F. Ndzebe-Ndoumba, Saskia D. Davi, Daniel Stelzl, Luzia Veletzky, Andrea Kreidenweiss, Tamara Nordmann, Ayola A. Adegnika, Bertrand Lell, Peter G. Kremsner, Michael Ramharter, Ghyslain Mombo-Ngoma.

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
