## [Decision Letter · Decision Letter 0]

19 Apr 2022

Dear Dr. MOMBO-NGOMA,

Thank you very much for submitting your manuscript "Effectiveness of antimalarial drug combinations in treating concomitant urogenital schistosomiasis in malaria patients in Lambaréné, Gabon: A non-randomised event-monitoring study." for consideration at PLOS Neglected Tropical Diseases. As with all papers reviewed by the journal, your manuscript was reviewed by members of the editorial board and by several independent reviewers. In light of the reviews (below this email), we would like to invite the resubmission of a significantly-revised version that takes into account the reviewers' comments. 

We appreciate the opportunity to review this interesting paper, which can be made stronger with attention to the reviewers' comments and further clarifications of the study design, egg quantification, group allocation, and follow-up.

We cannot make any decision about publication until we have seen the revised manuscript and your response to the reviewers' comments. Your revised manuscript is also likely to be sent to reviewers for further evaluation.

Sincerely,

Jennifer A. Downs, M.D., Ph.D.

Associate Editor

Amaya Bustinduy

Deputy Editor

We appreciate the opportunity to review this interesting paper, which can be made stronger with attention to the reviewers' comments and further clarifications of the study design, egg quantification, group allocation, and follow-up.

Reviewer's Responses to Questions

**Key Review Criteria Required for Acceptance?**

**Methods**

-Are the objectives of the study clearly articulated with a clear testable hypothesis stated?

-Is the study design appropriate to address the stated objectives?

-Is the population clearly described and appropriate for the hypothesis being tested?

-Is the sample size sufficient to ensure adequate power to address the hypothesis being tested?

-Were correct statistical analysis used to support conclusions?

-Are there concerns about ethical or regulatory requirements being met?

Reviewer #1: This is clear

Reviewer #2: Overall, the results presented are interesting. However, the data concerning artefenomel-ferroquine and artesunate-amodiaquine are not sufficiently relevant. They concern a small number of patients: older in the ASAQ group and too young in the OZFQ group compared to the AL and AP groups. I would suggest removing them.

In terms of methodology, The diagnostic aspects, in particular the method of determining the number of eggs excreted, must be clearly presented.

It is important to specify the method for determining the parasite load. Were all the eggs counted or a formula used? The readers of the article should find all this information in the methodology, this will allow a reproductibility of the used methodology. How was determined the egg viability at week 4 and week 6 ?

Were there patients not excluded for carriage of non-hematobium species or of hybrid Schistosoma sp whose carriage frequency in Gabon is not negligible? 

Although this is an observational study, patient selection criteria based on characteristics, such as the parasite load, should have been established because the difference in the median eggs count at baseline is significant between the groups (19 times higher in the AL group compared to the AP group). This has an impact on parasite clearance at week 4 and week 6. Similarly, the molecules studied differ in the combinations used. Authors should discuss this more explicitly. 

How were patients who did not participate in a clinical trial selected? , 

Were there any participants with macroscopic hematuria? if yes, what was the evolution of this hematuria after 6 weeks?

The subdivision of the groups for the endpoint analysis at week 4 and week 6 needs to be clarified

**Results**

-Does the analysis presented match the analysis plan?

-Are the results clearly and completely presented?

-Are the figures (Tables, Images) of sufficient quality for clarity?

Reviewer #1: This section is clear

Reviewer #2: The presentation of results must provide additional information to allow a better understanding of their observed effectiveness. Indeed, there were 24 patients included in the AP group, among them 16 and 19 were followed up at week 4 and week 6. However, the flow chart shows that 21 patients were included in the analysis. So there are 3 patients missing. In addition, the baseline data must relate to the 21 or 19 patients and not 24. The same applies to the AL group, the information provided in Table I relates to 23 patients, whereas the flow chart shows that only 18 were included in the analysis.

 In Table 2, the numbers at week-4 and those at week-6 differ; authors should explain why.

Regarding the egg counts in Table 2, it would be more accurate to determine the parasite load in those who still excrete eggs, by excluding those who do not. This would make it possible to better estimate the reduction in egg excretion, in those who were cured and even in patients who still carry eggs (was the egg count differents between the time points, especially in those still positives?). 

The information provided in the ASAQ group and the OZFQ group do not appear to be consistent. There is only 1 positive patient in the ASAQ group,so why are there ranges presented, as well as the two % reduct’ion in excretion? Regarding the OZFQ group, the baseline eggs excretion are different but the ranges are similar. 

The results in Table 2 suggest an absence of impact on egg excretion in the AL and OZFQ groups at week 4. It would be interesting to have information on the presence of other symptoms and their duration. The authors should mention it in the discussion, especially since the parasite densities are generally low, apart from the AL group.

Table 3. If none of the samples from patients with low egg count were positive at week 4 post-treatment, how do the authors explain the range of 0-37 and the % shedding of -23%-100%?

Were the differences observed between the groups statistically significant?

**Conclusions**

-Are the conclusions supported by the data presented?

-Are the limitations of analysis clearly described?

-Do the authors discuss how these data can be helpful to advance our understanding of the topic under study?

-Is public health relevance addressed?

Reviewer #1: Yes, this is clear

Reviewer #2: Although the interest of this study is undeniable, and the discussion well conducted, it addresses too succintly some important aspects.

The limitations of the study should be clearly discussed : participant selection method, the absence or presence of matching between the groups. and even the presentation of the data are major bias factors for the interpretation of the results obtained. The duration of the follow-up must be justified, eecause other authors report follow-up of up to several months with artemisinins and ACT.

The discussion is focused on excretion rates and parasite densities. The CERMEL team has been working for several years on schistosomiasis, the authors therefore have data on the parasite loads of Schistosoma haematobium in carriers. Are they comparable to those of the participants in this study? What about the efficacy of Praziquantel after 4 or 6 weeks follow-up ? There are several reports from Africa, including metaanalysis. It is important todiscuss the results obtained and those of PZQ regimen, even from other studies. 

Doesn't the fact of not having compared the groups after the administration of praziquantel constitute another bias? Several authors have shown the superiority of the association of artemisinin derivatives with praziquantel, compared to that obtained when they are administered alone.

It clearly appears that the molecules used have no action on adults, especially parous females. What do the authors think?

The absence of a significant effect in high parasite load should be taken into consideration when it is known that chronic carriage which is associated with high egg count is frequent in endemic regions.

Finally, it seeme that the duration of treatment with ACT alone is too short to obtain a good efficacy in reducing egg excretion. In the context of surveillance of resistance to artemisinin derivatives in Africa, what are the authors comments.

**Editorial and Data Presentation Modifications?**

Reviewer #1: See attached document

Reviewer #2: (No Response)

**Summary and General Comments**

Reviewer #1: See attached document

Reviewer #2: This observationaL study is an original contribution for the integrated management of coendemic parasitic diseases with high burden in Africa. It has been ethically well conducted as consenting participantswere screened and benefitted for treatment. 

Finding alternatives to praziquantel, which is still the only molecule recommended for the treatment of schistosomiasis, is a priority, especially in regions of high endemicity in which this NTD and its medical, psychological and socio-economic consequences are prevalent.

These are promising data in the context of the integrated approach to the management of endemic pathologies. Moreover, although the efficacy of artemisinin derivatives on schistomas has been studied for more than 10 years, such data on Schistosoma haematobium are scarce. 

Ethical standards are respected.

The article is quite well written. However, there are too long sentences with repetitive expressions and a considerable lack of punctuation, sometimes making comprehension difficult (see lines 113 to 120, 163 to 170, 293 to 295, 306 to 310). Punctuation should be corrected in lines 112, 113 to 117, the rest of the article should be checked. A verb is missing in lines 116 to 117. The end of the sentence at line 171 needs to be reformulated.

The description of the methodology used should be improved to allo the reproductibility. Some results need to be clarified and the statistical significance is missing; If the authors can perform the revision as required, this articule should be published.

PLOS authors have the option to publish the peer review history of their article (what does this mean?). If published, this will include your full peer review and any attached files.

Reviewer #1: Yes: Humphrey D. Mazigo

Reviewer #2: No
---

## [Decision Letter · Decision Letter 1]

17 Oct 2022

Dear Prof. MOMBO-NGOMA,

We are pleased to inform you that your manuscript 'Effectiveness of antimalarial drug combinations in treating concomitant urogenital schistosomiasis in malaria patients in Lambaréné, Gabon: A non-randomised event-monitoring study.' has been provisionally accepted for publication in PLOS Neglected Tropical Diseases.

We ask that you please do also consider the reviewer's note about further addressing the study limitations, as in the comment on the Results: "This is a secondary analysis and not planned at the start of the project. Authors need to acknowledge this."

Best regards,

Jennifer A. Downs, M.D., Ph.D.

Academic Editor

Amaya Bustinduy

Section Editor

Thank you for your attention to detail and your revisions, which we feel have significantly improved the manuscript.

Reviewer's Responses to Questions

**Key Review Criteria Required for Acceptance?**

**Methods**

-Are the objectives of the study clearly articulated with a clear testable hypothesis stated?

-Is the study design appropriate to address the stated objectives?

-Is the population clearly described and appropriate for the hypothesis being tested?

-Is the sample size sufficient to ensure adequate power to address the hypothesis being tested?

-Were correct statistical analysis used to support conclusions?

-Are there concerns about ethical or regulatory requirements being met?

Reviewer #1: All the comments have been addressed and the manuscript can be accepted

**Results**

-Does the analysis presented match the analysis plan?

-Are the results clearly and completely presented?

-Are the figures (Tables, Images) of sufficient quality for clarity?

Reviewer #1: This is a secondary analysis and not planned at the start of the project. Authors needs to acknowledge this, as it have results into low number of participants in come groups and has affected the analysis

**Conclusions**

-Are the conclusions supported by the data presented?

-Are the limitations of analysis clearly described?

-Do the authors discuss how these data can be helpful to advance our understanding of the topic under study?

-Is public health relevance addressed?

Reviewer #1: Yes

**Editorial and Data Presentation Modifications?**

Reviewer #1: Minor revision and accept for publication. I understand, there are limitation on this study but it still have data for further studies to assess their objectives

**Summary and General Comments**

Reviewer #1: The study offer another insight on the benefit of antimalarial as an alternative drug for schistosomiasis treatment. The findings support the earlier observations by other authors. Authors did not address the issue of variation in amount of active ingradients in various malaria drugs assessed in the study, which partly will address the issue of different efficacy observed

PLOS authors have the option to publish the peer review history of their article (what does this mean?). If published, this will include your full peer review and any attached files.

Reviewer #1: **Yes: **HUmphrey Deogratias Mazigo

---

## [Editor Report · Acceptance letter]

28 Oct 2022

Dear Prof. MOMBO-NGOMA,

We are delighted to inform you that your manuscript, "Effectiveness of antimalarial drug combinations in treating concomitant urogenital schistosomiasis in malaria patients in Lambaréné, Gabon: A non-randomised event-monitoring study.," has been formally accepted for publication in PLOS Neglected Tropical Diseases.

Best regards,

Shaden Kamhawi

co-Editor-in-Chief

Paul Brindley

co-Editor-in-Chief
